# Use of Complementary and Alternative Medicine in Patients with Rare Bone Diseases and Osteoporosis

**DOI:** 10.3390/nu16060816

**Published:** 2024-03-13

**Authors:** Roland Kocijan, Amadea Medibach, Lisa Lechner, Judith Haschka, Annemarie Kocijan, Daniel Arian Kraus, Jochen Zwerina, Martina Behanova

**Affiliations:** 1Ludwig Boltzmann Institute of Osteology at Hanusch Hospital of OEGK and AUVA, Trauma Centre Meidling, 1st Medical Department Hanusch Hospital, 1140 Vienna, Austria; amadea.medibach@icloud.com (A.M.); lisa.lechner@meduniwien.ac.at (L.L.); judith.haschka@osteologie.lbg.ac.at (J.H.); daniel.kraus@osteologie.lbg.ac.at (D.A.K.); jochen.zwerina@osteologie.lbg.ac.at (J.Z.); martina.behanova@osteologie.lbg.ac.at (M.B.); 2Vienna Bone and Growth Center, 1130 Vienna, Austria; 3Metabolic Bone Diseases Unit, School of Medicine, Sigmund Freud University Vienna, 1020 Vienna, Austria; 4Optimal Essen e.U, 1040 Vienna, Austria; kocijan@optimalessen.com

**Keywords:** rare bone disease, complementary and alternative medicine, CAM, osteoporosis, osteogenesis imperfecta, hypophosphatasia, X-linked hypophosphatemia

## Abstract

(1) Background: The use of complementary and alternative medicine (CAM) has seen a notable increase in popularity. However, there is an absence of data regarding the prevalence of CAM use in patients with rare bone diseases (RBDs). (2) Methods: This monocentric, cross-sectional study was carried out in a reference hospital for RBDs. RBD patients included individuals with osteogenesis imperfecta, hypophosphatasia and X-linked hypophosphatemia, and their data were compared with those of patients with osteoporosis (OPO) and of healthy controls (CON). This study utilized the German version (I-CAM-G) of the I-CAM questionnaire. (3) Results: This study comprised 50 RBD patients [mean age (SD) of 48.8 (±15.9), 26% male], 51 OPO patients [66.6 (±10.0), 9.8% male] and 52 controls [50.8 (±16.3), 26.9% male]. Treatments by naturopaths/healers were more prevalent in the RBD group (11.4%) compared with OPO (0%) and CON (5.8%) (*p* = 0.06). More than half of the OPO (60.8%) and CON (63.5%) patients and 46% of the RBD patients reported vitamin/mineral intake within the past 12 months (*p* = 0.16). Individuals with tertiary education had a significantly higher odds ratio of 2.64 (95% CI: 1.04–6.70, *p* = 0.04) for visiting any CAM provider. Further, OPO patients were significantly less likely to use self-help techniques compared with the CON group (OR = 0.42, 95% CI: 0.19–0.95; *p* = 0.04). (4) Conclusions: Herbal medicine, vitamin and mineral supplements, and self-help techniques were the most common forms of CAM reported by patients with RBDs. However, the use of CAM was generally low.

## 1. Introduction

Rare bone diseases (RBDs), such as osteogenesis imperfecta (OI), X-linked hypophosphatemia (XLH) and hypophosphatasia (HPP), are characterized by fractures, pseudo-fractures, bone deformities, musculoskeletal pain, impaired mobility and extraskeletal manifestations [1,2,3]. Consequently, the mental and, even more, physical quality of life (QoL) is reduced [4]. In recent years, specific therapies have been introduced for RBDs. Burosumab, a monoclonal FGF-23-antibody, is available for XLH. Studies have shown positive effects on bone mineralization, fracture healing and improvement in mobility [5]. For HPP, the enzyme replacement therapy asfotase alfa was approved for patients with pediatric-onset disease to treat bone manifestations. Asfotase alfa leads to bone mineralization, pseudofracture healing and improvement in physical function [6]. Although specific treatments are available, at least for XLH and HPP, these agents are currently reserved for severely affected patients. Moreover, besides the well-known positive effects on bone tissue, other clinical manifestations do not seem to be ameliorated.

To date, no specific drug is approved for OI. Thus, medical agents come from osteoporosis therapy. Bisphosphonates are known to increase bone mineral density in children and adults with OI [7]. However, fracture risk reduction in adult OI patients remains unclear [8]. Osteoanabolic agents such as teriparatide are treatment options for mild OI type I but show weaker results in moderate and severe forms [9]. Anti-sclerostin antibodies and TGF-beta inhibition could be therapeutic options in future [10]. Consequently, RBD patients suffer from numerous health issues, not covered by traditional medicine; employing coping mechanisms and resorting to alternative methods, therefore, seem the obvious choices in this population. Complementary and alternative medicine (CAM) is the terminology used to refer to medical methods that are not part of standard medical care. These include mind–body therapies, such as meditation, yoga and Tai Chi; biologically based practices, such as nutritional supplements and herbal medicine; and manipulative and body-oriented practices, such as manual therapy and acupuncture. There are also entire medical systems centered around CAM, such as traditional Chinese medicine and Ayurvedic medicine [11]. CAM utilization is well documented within oncology, where cancer patients seek relief in complimentary medicine besides evidence-based medicine [12]. Additionally, CAM use has been associated with quality of life in patients with chronic diseases. For example, a study indicated that CAM users with inflammatory bowel disease reported lower health-related quality of life (HRQoL), as measured by the SF-36, compared with non-users [13]. Similarly, patients with Type 2 Diabetes Mellitus who used some form of CAM also exhibited lower HRQoL (based on the EuroQoL tool) than those without any CAM use [14]. It is important to note that within the context of HRQoL measurements, a higher score is associated with a better health status and quality of life, suggesting that CAM use in these cases was associated with worse perceived health status and quality of life.

In Europe, CAM usage is more frequently reported by females than males. On the individual level, positive predictors of CAM use include female gender, higher socioeconomic status, longstanding illness, healthcare utilization, unmet medical needs and a negative opinion of the state of the health services [15]. Countries’ health expenditure is positively associated with the prevalence of CAM treatments; i.e., countries where health insurance offers more reimbursements for CAM present greater integration of such practices in the healthcare system [15]. A national survey in Norway, where CAM services are predominantly offered outside of the national healthcare service and at the patients’ expense, revealed that over half of the participants had used CAM in the last 12 months [16]. The prevalence of CAM use among patients with RBDs remains unclear.

Therefore, the aim of the present study was to assess differences in the prevalence and types of complementary and alternative medicine use among patients with RBDs, osteoporosis patients and healthy individuals. Additionally, we aimed to assess the association between CAM usage and socioeconomic status. We hypothesized that RBD patients utilize CAM more than osteoporosis patients and healthy controls and that individuals with higher socioeconomic status are more engaged in CAM.

## 2. Materials and Methods

This cross-sectional study was carried out at Hanusch Hospital, Vienna, a reference hospital for bone diseases and part of the Vienna Bone and Growth Center, Vienna, Austria (European Reference Network Center for Rare Bone Disease—ERN BOND). Data collection was performed between January 2021 and August 2022. Study participants were included after signing a written consent form. The study was approved by the Ethical Committee of the City of Vienna (EK 20-214-VK) on 10 November 2020 and was conducted in accordance with the Declaration of Helsinki. 

### 2.1. Study Groups

The rare bone disease group (RBD) consisted of adult patients with OI (Types I, III or IV according the classical Sillence Classification), XLH and HPP that had been genetically (or clinically) diagnosed during their routine visits. These data were compared to those of a group of male and female patients with osteoporosis (OPO) and a healthy control group (CON). Osteoporosis was defined by (i) T-scores ≤ −2.5 at the lumbar spine or hip, (ii) occurrence of major osteoporotic fractures (MOFs: hip, spine, forearm, humerus or pelvis) or (iii) bone structure defects and high fracture risk. Controls (CON) were bone-healthy subjects without any known history of osteoporosis or other metabolic bone disease. CON subjects were individuals who had either got surgery clearance for routine operations, inpatients (e.g., gynecological and ear–nose–throat operations) or volunteers from the general population (hospital staff, hospital visitors or family members). More details on study population including history of fractures and quality of life were recently published elsewhere [4]. 

Subjects were included in the study if they were ≥18 years old and able to read and understand the German I-CAM questionnaire.

### 2.2. I-CAM-G Questionnaire

The study utilized self-administered questionnaires for self-assessment. We used the German version (I-CAM-G) of the I-CAM-Q [17]. The use of the I-CAM-G in Austria is based on the validation and applicability of the German version [18]. 

The questionnaire contains questions focused on (i) CAM services, such as visits to alternative healthcare providers (homeopath, acupuncturist, herbalist, healer, osteopath, chiropractor); (ii) complementary treatments delivered by physicians in the field of naturopathy (homeopathy, acupuncture, herbal medicine, manual therapy or traditional Chinese medicine); (iii) use of CAM products (homeopathic remedies, herbal medicine or vitamins/minerals; in any form, prescribed and/or self-prescribed); (iv) self-help techniques or strategies (meditation, yoga, qigong, Tai Chi, relaxation technique, visualization, artistic activities or prayer).

Data on age, sex, marital status, highest educational level and employment status were also obtained by questionnaire. The educational level was categorized as basic (comprising primary education only), secondary (also comprising high school with and without leaving examination) and tertiary (also comprising university education) education. Marital status (single, married or cohabiting, divorced, or widowed), educational level and employment status (being employed or not) were utilized as indicators of socioeconomic status. 

The study design is shown in Figure 1.

### 2.3. Statistical Analysis

For describing the characteristics of the RBD, OPO and CON groups, frequencies and percentages were used for categorical variables. For continuous variables, the decision between using means and standard deviations (SDs) or medians and interquartile ranges (IQRs) was based on assessments of normality using the Shapiro–Wilk test. 

Differences in patient groups regarding demographic parameters were assessed by the Pearson chi-square test for categorical variables and the independent-samples Kruskal–Wallis test for continuous variables, after verifying the assumption of non-normal distribution for the latter.

The prevalence of rates CAM use and self-help practices across the three patient groups were examined using the Pearson chi-square test. To explore associations between selected demographic and socioeconomic factors and the utilization of CAM, univariate logistic regression models were employed, including factors such as sex, age, educational level, employment status, family status and patient type as independent variables. From these models, odds ratios with 95% confidence intervals were calculated to quantify the associations. Additionally, we examined the interaction of each of these variables by patient type. 

A two-sided *p*-value less than or equal to 0.05 was considered to indicate statistical significance. Data analysis was conducted using SPSS V29 (IBM Corp., Armonk, NY, USA).

## 3. Results

The study comprised 50 RBD patients [mean age (SD) of 48.8 (±15.9), 26% male], 51 OPO patients [66.6 (±10.0), 9.8% male] and 52 controls [50.8 (±16.3), 26.9% male]. Patients with osteoporosis were significantly older than RBD and control individuals (*p* < 0.001). The main characteristics of the study population are displayed in Table 1. 

### 3.1. Demographic Data

No statistically significant differences were observed among the groups in terms of sex, educational level, BMI or family status. Regarding employment status, 58% of the RBD patients reported being employed. This contrasts with a higher employment rate among the control group (78.8%), while only 41.5% of the osteoporosis patients were employed (Table 1).

### 3.2. I-CAM-G Questionnaire

The percentages of complementary and alternative medicine use during the last 12 months are shown in Table 2.

#### 3.2.1. Physicians and Other Therapists Who Have Treated the Patients with Naturopathic Treatments within the Last 12 Months

Treatments delivered by non-medically trained naturopaths/healers were more prevalent in the RBD group (11.4%) compared with OPO (0%) and CON (5.8%) (*p* = 0.06).

No differences among the three subgroups were found in terms of visits with homeopaths, acupuncturists, herbalists, osteopaths, chiropractors or other specialists. However, osteopathic therapy was the one chosen most frequently by OPO patients (12.2%) and the healthy control group (CON) (11.5%) but not RBD patients (7.0%) (*p* = 0.67). 

OPO patients underwent manipulation techniques on the musculoskeletal system performed by chiropractors and osteopaths and manual therapy performed by physicians numerically more often than the other two groups; however, this was not significant (Table 2).

#### 3.2.2. Treatments Delivered by Physicians in the Field of Naturopathy

The most common physician-prescribed CAM treatment was herbal medicines, used by 18.4% of RBD patients, 17.6% of healthy controls and 7.9% of OPO patients (*p* = 0.34). No RBD patients and only a few OPO (4.8% and 4.9%) and CON (4.0% and 4.0%) individuals received acupuncture and traditional Chinese medicine, respectively (*p* = 0.42 for both). 

The prevalence of prescribed homeopathy was low in all groups with no significant differences (*p* = 0.64). Manual therapy was reported by 17.1% of OPO patients and 15.7% of CON individuals but only 7.9% of RBD patients (*p* = 0.44) (Table 2). 

#### 3.2.3. Herbal Medicine and Dietary Supplements

The most self-reported natural remedies across all study groups were vitamins and minerals. More than half of the OPO patients (60.8%) and bone-healthy individuals of the control group (63.5%) and as many as 46% of RBD patients reported vitamin/mineral intake within the past 12 months (*p* = 0.16). 

The second most frequent natural remedies were herbal products in RBD patients (28.0%) and vitamin D in the OPO and CON groups (52.9% and 38.5%, respectively). In fact, vitamin D supplementation was the only natural remedy where a significant difference among the study groups could be found (*p* = 0.02), with the lowest prevalence among RBD patients (26.0%) (Table 2). 

Use of herbal products as personal CAM was more frequent for all study groups than herbal medicine CAM treatment delivered by physicians. Homeopathic remedies were not common among RBD patients (8.0%) and CON individuals (7.7%) within the last 12 months, but they were in OPO patients (19.6%) (Table 2). 

#### 3.2.4. Self-Help Practice

Relaxation techniques were carried out significantly more often by RBD patients (34.1%) than by other participants (OPO, 7.3%; CON, 27.5%; *p* = 0.01). There was a noticeable preference for meditation among RBD patients, with 26.7% choosing this method, compared with OPO (9.1%) and CON (16.0%). This trend was, however, not statistically significant (*p* = 0.09). 

Yoga was quite common in all three subgroups, without significant differences (*p* = 0.50). “Praying for health” and “Painting/playing a musical instrument for health”, respectively, were reported by more than 20% of participants in the RBD, OPO and CON groups (*p* = 0.75 and 0.56, respectively). Overall, less frequently used self-help practices within the last 12 months (≤10% in all groups) were qigong, Tai Chi and visualization (Table 3).

### 3.3. Demographic and Socioeconomic Factors and CAM

Individuals with tertiary education had a significantly higher odds ratio (2.64; 95% CI: 1.04–6.70, *p* = 0.04) for visiting any CAM provider in the last 12 months compared with the reference category (individuals with basic education). 

Similarly, for the recommendation of any CAM treatments from a doctor in the last 12 months, the odds ratio was higher for individuals with tertiary education (OR 2.39; 95% CI: 1.00–5.67, *p*-value = 0.05), suggesting a potential trend toward significance, indicating that higher education may be associated with a greater likelihood of receiving CAM treatments. 

The interaction of each demographic and socioeconomic indicator by patient type did not show statistical significance, indicating that the influence of these demographic and socioeconomic factors on CAM did not vary significantly across the three patient groups. For the use of any self-help technique in the last 12 months, patients with osteoporosis had an odds ratio of 0.42 (95% CI: 0.19–0.95; *p* = 0.04), indicating that patients with osteoporosis were significantly less likely to use self-help techniques compared with the control group (Table 4).

## 4. Discussion

This cross-sectional study assessed for the first time the prevalence of complementary and alternative medicine in patients with rare bone diseases and osteoporosis and healthy subjects. RBD patients suffer from numerous conditions and symptoms, and there is currently no cure available. Consequently, we expected a high proportion of CAM use in these patients in the pursuit of relief. Interestingly, aside from the notable exception of vitamins and minerals, which were frequently reported across all groups, the use of other CAM modalities was uncommon among individuals with rare bone diseases and osteoporosis and in the control group.

Complementary and alternative medicine are becoming increasingly popular [19]. However, there is still a lack of sound evidence and diverse sources of research data. The gap between research and practice in the field of alternative medicine is exacerbated by the public’s perception of these methods as unscientific or anti-medical [20]. Regardless of this, attitudes towards the use of CAM appear to be rather positive. Studies show that most people would opt for alternative treatment methods if they were confronted with chronic illnesses [21]. However, this is not consistent with the results of our study. Despite the previously reported reduced health-related quality of life in RBDs, especially regarding the physical components, compared with osteoporosis patients and healthy controls and despite severe fatigue being found in approximately one-third of RBD patients and one-fourth of osteoporotic patients, only a minority of patients consulted an alternative medicine provider [4]. 

Naturopathic and complementary treatments did not differ among the RBD, OPO and CON group and were generally rare. However, non-medical healers were consulted more frequently by RBD patients. The most physician-prescribed treatments were herbal medicines. Interestingly, the RBD group ranked highest in this category but also had the highest rate of independent use of herbal medicines. The potential benefits of various traditional Chinese medicine (TCM) remedies and Chinese herbal medicines on bone have been reported previously. These data suggested positive effects such as stimulation of bone formation, enhanced fracture healing, an increase in BMD and suppressed bone resorption [22,23]. Although 4400 acupuncture-certified physicians were registered in Austria as of October 2019 [24], the interest in this type of practice within our study groups was low. Less than 5% reported acupuncture treatments provided by a medical professional. In osteoporosis patients, however, positive effects have previously been observed, particularly in terms of pain relief [25].

A Canadian study found that 57% of OPO patients used CAM surveyed at three osteoporosis clinics [26]. The preferred therapeutic options were herbs, relaxation techniques, massage therapies and megavitamins for bone health. The study showed that CAM users tended to be younger, more educated and had a lower psychological quality of life [26]. Higher education was also associated with a greater likelihood of receiving CAM treatments in the present study. Our findings are consistent with the outcomes of a Europe-wide study on CAM use and sociodemographic determinants and a British systematic literature research study suggesting that CAM use is associated with a higher level of education [27,28]. Studies that have surveyed household income and level of education as separate parameters often show a more consistent correlation between CAM use and level of education than between CAM use and income [28,29]. Therefore, higher CAM use among the better educated could not solely be explained by higher household income and thus more financial means to spend on healthcare and CAM services. Income was not surveyed in our study and might remain a relevant confounding factor. It is worth noting that a large proportion of CAM services in Austria are not covered by the national insurance system, and patients have to pay out of pocket [30].

However, in patients with rare diseases, health literacy and information-seeking behavior are higher than in the general population, regardless of education [31,32]. The scarce literature and lack of valid information concerning CAM in RBDs might be one possible explanation for its reluctant use. 

Besides herbal medicines, preferences in personal CAM use were vitamins and minerals. Vitamin D supplementation was reported by 26% of RBD patients, 53% of OPO patients and 39% of CON individuals. Especially, the relatively low number of patients with osteoporosis is surprising, as vitamin D is an essential component of therapy. Vitamin D and calcium are recommended in osteoporosis patients, especially during specific treatment for osteoporosis. In particular, calcium supplements (in combination with vitamin D) are recommended for patients whose daily intake is below 800 mg [33]. In addition, vitamin D supplementation has been reported to reduce the risk of falls [34]. The combination of vitamin D and calcium, but not the single dose, could also decrease the risk of fractures [35]. While recommendations on calcium and vitamin D have been stated in all 27 observed clinical practice recommendations on osteoporosis by Ng et al., herbal medicine and acupuncture were only recommended by one study, which was published in China [36]. Within the three RBDs, the recommendations regarding supplementation of calcium and vitamin D differ.

In OI, a balanced diet was recommended not only to optimize body composition but also to support specific pharmacological therapy [37,38]. OI patients are likely to suffer from vitamin D deficiency, due to reduced mobility and thus fewer outdoor activities [39]. Consequently, supplementations of both calcium and vitamin D might be mandatory. However, body size has to be taken into account, and supplements should be adapted in patients with short stature. In contrast, XLH, formally known as vitamin D-resistant rickets or osteomalacia, is characterized by high FGF-23 levels and thus low 1.25 vitamin D and phosphate levels [2]. Consequently, inhibition of FGF-23 or the substitution of phosphate and 1.25 vitamin D are treatment options. Vitamin D (25-hydroxyvitamin D) might be substituted to reach normal ranges. However, calcium supplements are not recommended, due to possible side effects such as hypercalciuria and kidney stones [40]. HPP represents a heterogenous disease with various symptoms, partly also associated with calcium and phosphate intake. An association with musculoskeletal and neuropsychiatric symptoms was reported in patients with very low or very high calcium intake [41]. In contrast, a balanced nutritional phosphorus/calcium ratio improved some musculoskeletal issues in HPP [42]. Consequently, a balanced diet appears more favorable than calcium supplementation in HPP patients. With regards to vitamin D, a recently published study found no negative effects of vitamin D supplementation in HPP patients and even advocated vitamin D supplementation in HPP to prevent further mineralization disorders [43]. The results of the present study, therefore, highlight not only the importance of patient education regarding the essential supplements depending on the disease but also potential side effects of self-prescribed supplementation.

Lastly, the self-help section of our questionnaire, which assessed strategies such as yoga, meditation, qigong or praying, revealed that more RBD patients reported using self-help strategies than OPO individuals. There is substantial evidence that these techniques are beneficial to overall health. A bibliometric analysis of 886 studies on qigong reported that 97% of authors found positive effects on physical function, pain, mental health or quality of life [44]. Studies on Tai Chi reported benefits on overall health and fitness [45], and a meta-analysis also suggested a reduction in risk of falls [46]. A systematic review found that also mindfulness meditation can relieve chronic pain symptoms and depression and improve quality of life in many diseases. The field of self-help options is huge, and these may be beneficial for improving quality of life and to relieve symptoms, but research evidence is too scarce to draw firm conclusions [47].

The present study has several strengths and limitations. The major strength of our study is that we were able to include a variety of rare bone disease patients, osteoporosis patients and healthy controls, thereby allowing for including a broad perspective in our analysis. Next, the prevalence of CAM usage reported in the existing literature varies enormously regarding the included treatments and its definition. Therefore, to tackle this challenge, our study employed a standardized questionnaire to enable better comparability. This methodological choice ensures more consistent and reliable data on CAM usage, thus enhancing the validity of our findings in the context of existing research. A notable limitation of this study is the natural age discrepancy among the patient groups, with osteoporosis patients typically being older than those with RBDs. Another limitation is the potential self-report bias, which could lead to underreported or inaccurately reported information. Another limitation was the high proportion of female participants. While this gender imbalance could potentially skew the findings, it is important to consider the context of CAM usage patterns. Previous research has indicated that CAM usage is more prevalent among females [15]. The monocentric and cross-sectional design is another limitation, affecting the generalizability of our findings to wider populations.

## 5. Conclusions

Patients with rare bone diseases most commonly reported using herbal medicine, vitamin and mineral supplements, and self-help techniques as forms of complementary and alternative medicine. The usage rate in this population was low, however, which might be attributed to several factors: a lack of promotion of alternative treatment options, skepticism towards alternative approaches in Austria, a strong trust in conventional medicine or the non-reimbursement for CAM treatment by health insurance companies. Current evidence indicates a need for more research on complementary medicine in patients with rare bone diseases before recommending these non-pharmacological options.

## Figures and Tables

**Figure 1 nutrients-16-00816-f001:**
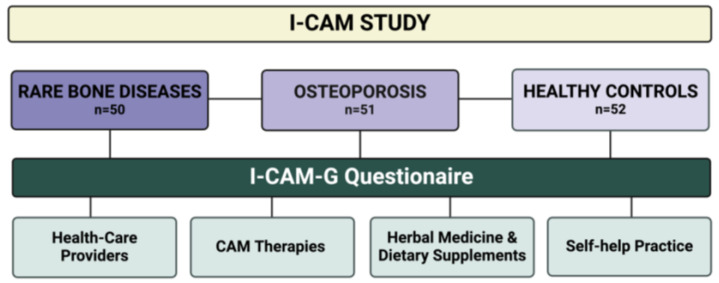
Study flow chart of the I-CAM study. The I-CAM-G questionnaire consists of 4 topics: (i) Healthcare providers: list of physicians and other therapists who have treated the patient with naturopathic treatments. (ii) CAM therapies: Treatments delivered by physicians in the field of naturopathy. (iii) Herbal medicine and dietary supplements: Use of herbal medicines and dietary supplements in any form (prescribed by a physician or therapist and all self-prescribed remedies). (iv) Self-help practice: Implementation or use of self-help techniques. I-CAM-G was available in the German language.

**Table 1 nutrients-16-00816-t001:** Basic characteristics of patient groups.

Patient Type	Rare Bone Disease Group (N = 50)				Osteoporosis Group (N = 51)	Control Group (N = 52)	
	OI (N = 17)	HPP (N = 17)	XLH (N = 16)	Total			*p*-Value *
**Age,** mean (SD)	47.6 (±15.6)	55.9 (±13.9)	42.5 (±16.0)	48.8 (±15.9)	66.6 (±10.0)	50.8 (±16.3)	**<0.001**
**Gender, male,** N (%)	5 (29.4)	7 (41.2)	1 (6.3)	13 (26.0)	5 (9.8)	14 (26.9)	0.06
**Family status,** N (%)							0.09
Single	4 (23.5)	1 (5.9)	3 (18.8)	8 (16.0)	4 (7.8)	13 (25.0)	
Married or cohabiting	9 (52.9)	10 (58.8)	9 (56.3)	28 (56.0)	25 (49.0)	30 (57.7)	
Divorced	2 (11.8)	3 (17.6)	4 (25.0)	9 (18)	13 (25.5)	5 (9.6)	
Widowed	0 (0.0)	1 (5.9)	0 (0.0)	1 (2.0)	5 (9.8)	4 (7.7)	
**Educational level,** N (%)							0.07
Basic	9 (52.9)	7 (41.2)	10 (62.5)	26 (52.0)	19 (37.3)	28 (53.8)	
Secondary	3 (17.6)	2 (11.8)	0 (0.0)	5 (10.0)	16 (31.4)	8 (15.4)	
Tertiary	3 (17.6)	6 (35.3)	6 (37.5)	15 (30.0)	13 (25.5)	16 (30.8)	
**Employment status, employed,** N (%)	9 (52.9)	8 (47.1)	12 (75.0)	29 (58.0)	22 (43.1)	41 (78.8)	**<0.001**
**BMI**	25.4 (±6.2)	27.2 (±5.1)	25.8 (±5.7)	26.2 (±5.6)	24.2(±3.9)	26.4 (±4.7)	0.16

RBD, rare bone disease; OI, osteogenesis imperfecta; HPP, hypophosphatasia; XLH, x-linked hypophosphatemia; BMI, body mass index; * *p*-value for group differences among RBD, OPO and CON groups. Statistically significant *p*-values are marked in bold. Missing values: Family status: RBD, 4 (8%); OPO, 4 (7.8%); CON, 0 (0%). Educational level: RBD, 4 (8%); OPO, 3 (5.9%); CON, 0 (0%). Employment status: RBD, 4 (8.0%); OPO, 1 (2.0%); CON, 0 (0%). BMI: RBD, 4 (8%); OPO, 4 (7.8%); CON, 7 (13.5%). Data are expressed as percentages for categorical variables and means and ± standard deviations for continuous variables.

**Table 2 nutrients-16-00816-t002:** Percentages of complementary and alternative medicine use during the last 12 months in the three patient groups.

	Any Treatment from Any Provider		CAM Treatment from Physicians		Natural Remedies	
	Provider, N (%)	*p*-Value	Treatment, N (%)	*p*-Value	Type of Remedy, N (%)	*p*-Value
	**Homeopath**	0.62	**Homeopathy**	0.64	**Homeopathic remedy**	0.10
RBD	1/45 (2.2)		1/39 (2.6)		4/50 (8.0)	
OPO	3/49 (6.1)		3/42 (7.1)		10/51 (19.6)	
CON	3/52 (5.8)		3/51 (5.9)		4/52 (7.7)	
	**Acupuncturist**	0.85	**Acupuncture**	0.42	**Herbal products**	0.31
RBD	1/45 (2.2)		0/38 (0.0)		14/50 (28.0)	
OPO	2/47 (4.3)		2/42 (4.8)		8/51 (15.7)	
CON	2/52 (3.8)		2/50 (4.0)		13/52 (25.0)	
	**Natural healer,** **MD (herbalist)**	0.85	**Herbal medicine**	0.34	**Vitamins/minerals**	0.16
RBD	2/44 (4.5)		7/38 (18.4)		23/50 (46.0)	
OPO	3/48 (6.3)		3/38 (7.9)		31/51 (60.8)	
CON	2/52 (3.8)		9/51 (17.6)		33/52 (63.5)	
	**Naturopath,** **non-MD (healer)**	0.06	**Manual therapy**	0.44	**Vitamin D**	0.02
RBD	5/44 (11.4)		3/38 (7.9)		13/50 (26.0)	
OPO	0/46 (0.0)		7/41 (17.1)		27/51 (52.9)	
CON	3/52 (5.8)		8/51 (15.7)		20/52 (38.5)	
	**Osteopath**	0.67	**Traditional Chinese medicine**	0.42	**Other remedies**	0.72
RBD	3/43 (7.0)		0/37 (0.0)		3/50 (6.0)	
OPO	6/49 (12.2)		2/41 (4.9)		2/51 (3.9)	
CON	6/52 (11.5)		2/50 (4.0)		4/52 (7.7)	
	**Chiropractor**	0.25				
RBD	0/43 (0.0)					
OPO	3/46 (6.5)					
CON	3/51 (5.9)					
	**Other specialists**	0.97				
RBD	2/38 (5.3)					
OPO	2/39 (5.1)					
CON	3/49 (6.1)					

CAM, complementary and alternative medicine; RBD, rare bone disease group; OPO, osteoporosis group; CON, healthy control group. Level of significance *p* ≤ 0.05.

**Table 3 nutrients-16-00816-t003:** Prevalence of self-help practices used in the last 12 months by patients with RBD or osteoporosis and healthy controls.

	Prevalence by Patient Group N (%)			*p*-Value
	RBD	OPO	CON	
**Meditation**	12/45 (26.7)	4/44 (9.1)	8/50 (16.0)	0.09
**Yoga**	9/45 (20.0)	6/44 (13.6)	12/52 (23.1)	0.50
**Qigong**	0/42 (0.0)	1/42 (2.4)	2/50 (4.0)	0.43
**Tai Chi**	1/43 (2.3)	1/41 (2.4)	1/50 (2.0)	0.99
**Relaxation techniques**	15/44 (34.1)	3/41 (7.3)	14/51 (27.5)	0.01
**Visualization**	4/43 (9.3)	1/41 (2.4)	5/50 (10.0)	0.34
**Praying for health**	10/43 (23.3)	9/42 (21.4)	14/50 (28.0)	0.75
**Painting/playing a musical instrument for health**	11/44 (25.0)	13/44 (29.5)	10/50 (20.0)	0.56
**Other techniques**	4/22 (18.2)	8/28 (28.6)	20/40 (50.0)	0.03

RBD, rare bone disease group; OPO, osteoporosis; CON, healthy controls; level of significance *p* ≤ 0.05.

**Table 4 nutrients-16-00816-t004:** Associations between selected demographic and socioeconomic factors and CAM utilization in the last 12 months assessed by univariate logistic regression represented by odds ratios and 95% confidence intervals (ORs and 95% CIs).

Outcome	Visit with Any CAM Provider in the Last 12 Months		Having Received any CAM Treatment from a Doctor in the Last 12 Months		Use of Any Self-Help Technique in the Last 12 Months	
	OR (95% CI)	*p*-Value	OR (95% CI)	*p*-Value	OR (95% CI)	*p*-Value
**Sex**						
Male	1 (Reference)		1 (Reference)		1 (Reference)	
Female	1.07 (0.40–2.90)	0.89	1.50 (0.56–3.96)	0.42	1.75 (0.80–3.84)	0.16
**Age category**						
<60	1.21 (0.54–2.73)	0.64	0.89 (0.42–1.87)	0.76	1.21 (0.63–2.33)	0.56
≥60	1 (Reference)		1 (Reference)		1 (Reference)	
**Patient type**						
RBD	0.44 (0.16–1.21)	0.11	0.49 (0.19–1.25)	0.14	0.61 (0.27–1.39)	0.32
Osteoporosis	0.58 (0.23–1.50)	0.26	0.69 (0.29–1.66)	0.41	0.42 (0.19–0.95)	0.04
Controls	1 (Reference)		1 (Reference)		1 (Reference)	
**Education**						
Basic	1 (Reference)		1 (Reference)		1 (Reference)	
Secondary	2.00 (0.68–5.91)	0.21	2.08 (0.77–5.58)	0.15	1.40 (0.57–3.43)	0.46
Tertiary	2.64 (1.04–6.70)	0.04	2.39 (1.00–5.67)	0.049	1.76 (0.79–3.90)	0.16
**Employment**						
Not employed	1 (Reference)		1 (Reference)		1 (Reference)	
Employed	2.33 (0.93–5.87)	0.07	1.61 (0.72–3.59)	0.24	0.84 (0.43–1.72)	0.68
**Family status**						
Single	1.88 (0.32–10.97)	0.48	0.58 (0.12–2.71)	0.49	0.91 (0.18–4.47)	0.91
Married or cohabiting	0.95 (0.18–4.94)	0.96	0.41 (0.11–1.63)	0.21	0.65 (0.16–2.69)	0.65
Divorced	0.70 (0.11–4.55)	0.71	0.52 (0.11–2.42)	0.41	0.86 (0.18–4.13)	0.85
Widowed	1 (Reference)		1 (Reference)		1 (Reference)	

## Data Availability

The datasets generated and analyzed during the current study are not publicly available due to the risk of indirect identification, where pseudonymized data could inadvertently reveal participants’ identities when cross-referenced with other public information, but they are available from the corresponding author upon reasonable request.

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
