# Peer review of "Use of Complementary and Alternative Medicine in Patients with Rare Bone Diseases and Osteoporosis"

_nutrients, 2024, doi:10.3390/nu16060816_

Round 1

Reviewer 1 Report

Comments and Suggestions for Authors

The manuscript entitled “ The Use of Complementary and Alternative Medicine in Patients with Rare Bone Diseases” deals with cross-sectional studies of the use of complementary and alternative medicine by patients with rare bone diseases. Survey results showed that complementary and alternative medicine is not popular among these patients with the exception of herbal medicine, vitamin and mineral supplementation and self-improvement techniques. Results were discussed in reference to other articles.

Cited references are quite new.

Statistical analysis was applied for characterization the obtained results.

Manuscript is well-written and contains valuable information. It is suitable for readers of Nutrients.

Author Response

Authors’ response: Thank you for taking time in reviewing our manuscript and for your positive feedback.

Reviewer 2 Report

Comments and Suggestions for Authors

In this study, the authors compared CAM usage of different bone disease such as RBD, osteoporosis and compared to control. The cross-sectional study is overall informative and writing is overall good and presentation is clear. However, it is still not clear which CAM are beneficial for RBD and osteoporosis except vitamins/mineral and vitamin D. Following are some of the comments:

1.       Title: not clear.  This needs to be more specific, authors included not only RBD but a common bone disease osteoporosis.

2.       Whether these CAM are beneficial for the RBD, Osteoporosis should be included. The questionnaire should include whether these CAM treatment are beneficial for each group of patients. Specific CAM’s benefit should be included instead of just looking at the usage. If there is no benefit, choosing these CAM will be low. RBD is genetic disease, these CAM may not be useful except self-help techniques.

3.       Line 67 to 69 “For example, a study indicated that CAM users with inflammatory bowel disease reported lower health-related quality of life (HRQoL), as measured by the SF-36, compared to non-users (13). Similarly, patients with Type 2 Diabetes Mellitus who used some CAM also exhibited lower HRQoL, (based on the EuroQoL tool) than those without any CAM use (14).”   Authors should state whether lower HEQoL is better or higher is better.  It is confusing for non-clinical reader. CAM is associated with lower HRQoL means lower is better?

4.       Line 230-234 Odds ratio is higher for both CAM doctor visit or treatment in tertiary education population, authors should explain higher odds ratio means more likely to seek these treatments in this case to avoid confusion.  In general, for case control study, Odds ratio high means high risk.  Odds ratio calculation should be mentioned in methods section instead of just mention the software.

5.       Table 2 Said “prevalence” of each CAM usage. However, this word is for a disease, but not medical practices, may be using frequency or percentage.

6.       Line 253 “Interestingly, the use of CAM was uncommon in RBD, OPO and CONs, respectively? What does the sentence mean? Are CAM uncommon for all Groups, then why authors said respectively? Vitamin/mineral use in the current study is quite high.

7.       Line 276, “reference 24” position is not correct. Please correct.

Author Response

#Reviewer 2

Comments and Suggestions for Authors

In this study, the authors compared CAM usage of different bone disease such as RBD, osteoporosis and compared to control. The cross-sectional study is overall informative and writing is overall good and presentation is clear. However, it is still not clear which CAM are beneficial for RBD and osteoporosis except vitamins/mineral and vitamin D. Following are some of the comments:

#1. Title: not clear.  This needs to be more specific, authors included not only RBD but a common bone disease osteoporosis.

Authors’ response: Thank you for pointing this to our attention. We changed the title as follows: “The Use of Complementary and Alternative Medicine in Patients with Rare Bone Diseases and Osteoporosis”.

#2. Whether these CAM are beneficial for the RBD, Osteoporosis should be included. The questionnaire should include whether these CAM treatment are beneficial for each group of patients. Specific CAM’s benefit should be included instead of just looking at the usage. If there is no benefit, choosing these CAM will be low. RBD is genetic disease, these CAM may not be useful except self-help techniques.

Authors’ response: Thank you for your insightful comments. We appreciate your suggestion to include questions related to the specific benefits of CAM treatments for these patient groups in our questionnaire. We utilized a standardized questionnaire to facilitate comparability and consistency across different studies and populations and we did not aim to modify it.

While we acknowledge the importance of assessing the benefits of CAM treatments, such an investigation would require a more detailed and condition-specific approach, potentially involving clinical trials or longitudinal studies to accurately measure outcomes and benefits. This aspect was beyond the scope of our current research, which aimed to provide a preliminary overview of CAM usage patterns among individuals with RBD and osteoporosis.

We would like to clarify that the primary aim of our study was to investigate the usage patterns of CAM among individuals with with RBD and osteoporosis and bone healthy controls. This focus was informed by existing literature reviews, which demonstrated a significant interest in CAM usage among patients with chronic conditions, including cancer. These reviews indicated that understanding the prevalence and types of CAM used could provide valuable insights into patient preferences and healthcare behaviors in these populations.

We believe that our findings will contribute to a growing body of knowledge regarding CAM usage in chronic conditions and potentially inform future research focused on evaluating the efficacy and benefits of specific CAM treatments for these patient groups.

#3. Line 67 to 69 “For example, a study indicated that CAM users with inflammatory bowel disease reported lower health-related quality of life (HRQoL), as measured by the SF-36, compared to non-users (13). Similarly, patients with Type 2 Diabetes Mellitus who used some CAM also exhibited lower HRQoL, (based on the EuroQoL tool) than those without any CAM use (14).”   Authors should state whether lower HEQoL is better or higher is better.  It is confusing for non-clinical reader. CAM is associated with lower HRQoL means lower is better?

Authors’ response: Thank you for your careful reading of our manuscript and for pointing out the need for clarification on the interpretation of health-related quality of life (HRQoL). In the context of our manuscript, and generally in health-related quality of life research, a higher HRQoL score is indicative of a better health status and quality of life. Therefore, when we report that CAM users with inflammatory bowel disease and patients with Type 2 Diabetes Mellitus exhibited lower HRQoL scores compared to non-users, it implies that their perceived health status and quality of life were worse than those who did not use CAM.

To address this confusion, we revised the mentioned lines as follows:

"For example, a study indicated that CAM users with inflammatory bowel disease reported lower health-related quality of life (HRQoL), as measured by the SF-36, compared to non-users (13). Similarly, patients with Type 2 Diabetes Mellitus who used some forms of CAM also exhibited lower HRQoL, (based on the EuroQoL tool) than those without any CAM use (14). It is important to note that within the context of HRQoL measurements, a higher score is associated with a better health status and quality of life, suggesting that CAM use in these cases was associated with worse perceived health status and quality of life." (Lines 71-74)

#4. Line 230-234 Odds ratio is higher for both CAM doctor visit or treatment in tertiary education population, authors should explain higher odds ratio means more likely to seek these treatments in this case to avoid confusion.  In general, for case control study, Odds ratio high means high risk.  Odds ratio calculation should be mentioned in methods section instead of just mention the software.

Authors’ response:

Thank you for your insightful comments and for drawing attention to the need for clarity in the interpretation of the odds ratios reported in our study.

In the context of our study, a higher odds ratio indicates a greater likelihood or propensity of individuals within a certain group (individuals with tertiary education) to seek out or use CAM treatments or services, rather than a risk factor for a negative health outcome.

To clarify this, we added:

“Individuals with tertiary education had a significantly higher odds ratio of 2.64 (95% CI: 1.04-6.70, p=0.04) for visiting any CAM provider in the last 12 months compared to the reference category (basic education). This indicates that among our sample, those with tertiary education were more likely to have visited a CAM provider compared to those with basic education.” (Line 246-248)

“Similarly, for the recommendation of any CAM treatments from a doctor in the last 12 months, for individuals with tertiary education the odds ratio was higher (OR 2.39; 95% CI: 1.00-5.67, p-value=0.05), suggesting a potential trend toward significance, indicating that higher education may be associated with a greater likelihood of receiving CAM treatments.” (Lines 251-253)

We also clarified the methods section:

To explore associations between selected demographic and socioeconomic factors and the utilization of CAM, univariate logistic regression models were employed, including factors such as sex, age, educational level, employment status, family status and patient type as independent variables. From these models, odds ratios with 95% confidence intervals were calculated to quantify the associations.”(Lines 159-161)

#5.Table 2 Said “prevalence” of each CAM usage. However, this word is for a disease, but not medical practices, may be using frequency or percentage.

Authors’ response: We modified the title of Table 2: Percentage of complementary and alternative medicine use during the last 12 months in three patients’ groups.” (Line 185)

#6. Line 253 “Interestingly, the use of CAM was uncommon in RBD, OPO and CONs, respectively? What does the sentence mean? Are CAM uncommon for all Groups, then why authors said respectively? Vitamin/mineral use in the current study is quite high.

Authors’ response:

Thank you for your insightful query. In our study, when we mention that the use of Complementary and Alternative Medicine (CAM) was uncommon in patients with Rare Bone Disease (RBD), Osteoporosis (OPO), and among the Control group (CONs), our intention was to highlight a general trend observed across all three groups. The term "respectively" was used to indicate that this observation applies to each group individually, reinforcing the consistency of low CAM usage across the different cohorts examined in our study.

However, we recognize that the use of "respectively" in this context might have unintentionally introduced ambiguity. To clarify, our analysis revealed that, broadly speaking, the utilization of CAM modalities—excluding vitamins and minerals—was indeed low across all three groups studied: RBD, OPO, and CONs. This indicates a general reluctance or lack of preference for CAM therapies among participants, regardless of their group classification.

For a more clarity, we revised the particular sentence as follows: “Interestingly, aside from the notable exception of vitamins and minerals, which were frequently reported across all groups, the use of other CAM modalities was uncommon among individuals with rare bone disease, osteoporosis and in the control group.” (Lines 272-275)

We hope this clarification addresses your concern and elucidates the findings presented in our study.

#7. Line 276, “reference 24” position is not correct. Please correct.

Authors’ response: We corrected the positioning of “reference 24”.

Reviewer 3 Report

Comments and Suggestions for Authors

Dear Authors,

Complementary and alternative medicine is used quite widely in society, however the specialist suggestions about significance of it is quite countrary. Despite this the authors addressed the research questions objectively and impersonally. The manuscript cant be revealed as the most unique and bringing a lot of novelty in the field, but the scientific design is acceptable, the obtained data comparable and results are followed by really interesting Discussion. Thus, I think that we need such a manuscripts in the medicine field just to give provable data for the readers and those who take the strict side of  either traditional or conventional medicine.

The methodology, results, discussion, limitations, conclusions are appropriate, easy readable. Also References are OK.

I would like just to include one small comment - please, add in the part 2 the research time (from to) and also add the Ethical Committee office and year of issue. Also inclusion criteria for the groups should be defined more clearly, the described groups remind a little mess now...

Otherwise, thank you, interesting research.

Round 2

Reviewer 2 Report

Comments and Suggestions for Authors

The authors have addressed most of my questions. It is now acceptable to be published in nutrients. Congratulations!